# Auraptene Mitigates Parkinson’s Disease-Like Behavior by Protecting Inhibition of Mitochondrial Respiration and Scavenging Reactive Oxygen Species

**DOI:** 10.3390/ijms20143409

**Published:** 2019-07-11

**Authors:** Yunseon Jang, Hyosun Choo, Min Joung Lee, Jeongsu Han, Soo Jeong Kim, Xianshu Ju, Jianchen Cui, Yu Lim Lee, Min Jeong Ryu, Eung Seok Oh, Song-Yi Choi, Woosuk Chung, Gi Ryang Kweon, Jun Young Heo

**Affiliations:** 1Department of Biochemistry, Chungnam National University School of Medicine, Daejeon 35015, Korea; 2Department of Medical Science, Chungnam National University School of Medicine, Daejeon 35015, Korea; 3Infection Control Convergence Research Center, Chungnam National University School of Medicine, Daejeon 35015, Korea; 4Research Institute for Medical Science, Chungnam National University School of Medicine, Daejeon 35015, Korea; 5Department of Neurology, Chungnam National University Hospital, Daejeon 35015, Korea; 6Department of Pathology, Chungnam National University School of Medicine, Daejeon 35015, Korea; 7Department of Anesthesiology and pain medicine, Chungnam National University Hospital, Daejeon 35015, Korea; 8Department of Anesthesiology and pain medicine, Chungnam National University, Daejeon 35015, Korea; 9Brain Research Institute, Chungnam National University School of Medicine, Daejeon 35015, Korea

**Keywords:** auraptene, dopamine neuron, Parkinson’s disease, neuroprotection, antioxidant, mitochondria

## Abstract

Current therapeutics for Parkinson’s disease (PD) are only effective in providing relief of symptoms such as rigidity, tremors and bradykinesia, and do not exert disease-modifying effects by directly modulating mitochondrial function. Here, we investigated auraptene (AUR) as a potent therapeutic reagent that specifically protects neurotoxin-induced reduction of mitochondrial respiration and inhibits reactive oxygen species (ROS) generation. Further, we explored the mechanism and potency of AUR in protecting dopaminergic neurons. Treatment with AUR significantly increased the viability of substantia nigra (SN)-derived SN4741 embryonic dopaminergic neuronal cells and reduced rotenone-induced mitochondrial ROS production. By inducing antioxidant enzymes AUR treatment also increased oxygen consumption rate. These results indicate that AUR exerts a protective effect against rotenone-induced mitochondrial oxidative damage. We further assessed AUR effects in vivo, investigating tyrosine hydroxylase (TH) expression in the striatum and substantia nigra of MPTP-induced PD model mice and behavioral changes after injection of AUR. AUR treatment improved movement, consistent with the observed increase in the number of dopaminergic neurons in the substantia nigra. These results demonstrate that AUR targets dual pathogenic mechanisms, enhancing mitochondrial respiration and attenuating ROS production, suggesting that the preventative potential of this natural compound could lead to improvement in PD-related neurobiological changes.

## 1. Introduction

Current therapeutics for Parkinson’s disease (PD) lack neuroprotective properties and are only effective in providing symptom relief [1]. To overcome the limitations of PD drugs, researchers have focused on early pathological changes in PD [1,2,3], with the goal of developing strategies for early interventions, prior to the onset of severe motor symptoms, such as bradykinesia, rigidity and resting tremors, in patients with preclinical or prodromal stage PD [4].

Oxidative stress on dopaminergic neurons causes neurodegeneration and induces behavioral symptoms of PD. More than 90% of intracellular reactive oxygen species (ROS) are produced by aberrant electron transfer during mitochondrial respiration [5,6]. There is some evidence to suggest that mitochondrial alterations lead to PD-like pathologies. For example, genetic mutations in the PD-related genes, *Parkin, DJ-1* or *PTEN-induced kinase 1*
*(**PINK1)*, cause mitochondrial dysfunction in offspring of familial-type PD patients, and 1-methyl-4-phenyl-1,2,3,6-tetrahydropyridine (MPTP) and rotenone, which are known to be PD-inducing toxins, inhibit mitochondrial complex I [6]. These two neurotoxins are suitable to show the effects of auraptene (AUR) in PD models, which results from mitochondrial dysfunction because both toxins lead to PD by inducing oxidative stress. The accumulation of α-synuclein, which has neurotoxic effects prior to the onset of PD symptoms, can also cause mitochondrial alterations and ROS production [7]. Therefore, modulating mitochondrial function during the pathogenesis of PD could be an effective preventive therapeutic strategy in prodromal stage PD.

Auraptene (AUR) is a 7-geranyloxylated coumarin isolated from citrus fruit [8]. Natural compounds such as AUR might generally be expected to offer advantages of safety and minimal adverse effects [9]; notably, AUR is able to cross the blood-brain barrier [10]. We previously showed that AUR inhibits progression of renal cell carcinoma by altering mitochondrial metabolism [11]. In addition to its anticancer effects, AUR has been used in conjunction with various toxins, including N-methyl-D-aspartate, lipopolysaccharide (LPS) and scopolamine, to study the neuroprotective effects of AUR against various neurotoxic defects (e.g., cerebral ischemia and neurodegenerative diseases), focusing on movement disorders and memory impairments [12,13,14,15]. Although AUR treatment inhibits microglial activation and prevents dopaminergic neuronal loss in an LPS mouse model [14], the molecular and cellular mechanisms for the protective effects of AUR in PD models are not yet clear, and the effects of AUR on motor function in PD have not yet been investigated.

In the context of cancer, biosynthetic substrates and energy supplied by mitochondria support cancer cell proliferation and metastasis. Because AUR treatment suppresses mitochondrial function, it leads to inhibition of cancer proliferation. However, in the context of neurodegeneration, maintenance or protection of neurotoxin-induced reduction in mitochondrial respiration increases neuronal activity and survival. In order to clarify the antioxidative effect by treatment with AUR, we investigated the alteration of cell viability, antioxidant enzyme expression and ROS generation by using rotenone, MPP^+^ in SN4741 cell line. We demonstrated that pretreatment with AUR improves movement deficits in association with an increase in the number of dopamine neurons in the substantia nigra (SN) of MPTP-induced PD mouse models which inhibits the mitochondrial complex I. On the basis of these findings, we suggest that AUR pretreatment acts through protection of a decrease in mitochondrial respiration by neurotoxins and down-regulation of ROS of dopaminergic neurons to produce its beneficial PD-related neurobiological changes.

## 2. Results

### 2.1. AUR Increases Cell Viability and Protects Against Neurotoxin-Induced Inhibition of Mitochondrial Respiration

Rotenone and 1-methyl-4-phenylpyridinium (MPP^+^), the active metabolite of MPTP, are commonly used neurotoxins in PD models [16,17]. Accordingly, we examined the protective effect of AUR on neurotoxin-induced cell death in dopaminergic neuron-like SN4741 cells. Using sulforhodamine B (SRB) assays to assess the viability of SN4741 cells after rotenone or MPP^+^ treatment, we found that these toxins caused cell death in a dose-dependent manner (Figure 1A,B). Notably, AUR pretreated SN4741 cells were resistant to the neurotoxicity of both rotenone and MPP^+^ compared to cells without AUR treatment (Figure 1A,B). At a concentration of 1 μM, AUR alone had no effect on cell viability, as shown in Appendix A.

It has previously been reported that AUR affects mitochondrial complex I and inhibits mitochondrial respiration in RCC4 renal cell carcinoma cells [8,11]. In this context, effects of AUR on mitochondrial oxygen consumption rate (OCR), shown in Figure 1C,D, are somewhat counterintuitive. In these experiments SN4741 cells were pretreated with AUR and then incubated with 0.25 μM rotenone for 24 h, after which the effects of AUR on mitochondrial respiration were determined by measuring OCR using an XF24 analyzer. Incubation with rotenone alone for 24 h led to a 67.8% reduction in the basal OCR area under the curve compared with that of the control group. Notably, treatment with AUR prior to rotenone treatment attenuated these effects, blunting the inhibitory effect of rotenone by 24.1% (Figure 1C,D). AUR alone and short-term cotreatment with AUR and rotenone did not change basal OCR level (Appendix A). Similar results were obtained following MPP^+^ treatment. The group treated with MPP^+^ only exhibited a 17% decrease in basal OCR (Figure 1E), whereas the AUR pretreated group showed a basal OCR that was 14.2% higher than that of controls (Figure 1F). Extracellular acidification rate (ECAR) was also increased in the AUR pretreated, rotenone-exposed group compared with the rotenone-only group, but was unchanged in the MPP^+^ group (Appendix A). Taken together, these results suggest that AUR protects against decreases in cell viability and suppression of mitochondrial respiration induced by neurotoxins in dopaminergic neuronal cells. 

### 2.2. AUR Induces Antioxidant Enzyme Expression in a Rotenone-Treated Cell Model

Antioxidant compounds protect against cellular responses to ROS, which cause oxidative cellular damage in PD [18,19,20,21,22,23]. Given previously reported antioxidant effects of AUR on lymphocytes treated with H_2_O_2_ [24], we hypothesized that AUR affects antioxidant enzyme expression in dopaminergic neuronal cells. As a first step in determining the effect of AUR on antioxidant systems, we measured the levels of NRF2 (nuclear factor, erythroid 2 like 2), a transcription factor inducing antioxidant-related gene [25] in SN4741 cells. We observed that NRF2 protein levels were significantly increased in rotenone or MPP^+^-treated cells pretreated with AUR compared with those in cells treated with either neurotoxin alone (Figure 2A–D). These results indicate that AUR treatment induces NRF2 protein expression in cells.

To determine whether AUR alters expression of ROS scavengers, we quantified the expression of transcripts of genes encoding antioxidant enzymes and those involved in glutathione (GSH) production and recycling using quantitative reverse transcription-polymerase chain reaction (RT-qPCR) [23,26,27]. Specifically, we analyzed transcript levels of Nrf2, Nqo1, Gpx1, Gst, Gclc, Gclm and Gr, as well as transcript levels of mitochondrial antioxidant enzymes, including Sod1 and Sod2. Nrf2, Nqo1, and Gpx1 mRNA levels were increased in AUR pretreated cells subsequently treated with rotenone or MPP^+^ (Figure 2E,F). In the case of enzymes involved in GSH production and regeneration, Gclc mRNA was induced by AUR in the presence of MPP^+^, but not in the presence of rotenone (Figure 2G,H). In SN4741 cells incubated in the presence of MPP^+^ for 24 h, both Sod2 mRNA and protein levels were comparable to those of controls, regardless of AUR pretreatment (Appendix A). Taken together, these results suggest that AUR prevents neurotoxin-induced oxidative damage in dopaminergic neurons by enhancing antioxidant enzyme expression. 

### 2.3. AUR Inhibits Rotenone-Induced Cytosolic ROS Production

Rotenone induces ROS production by inhibiting mitochondrial complex I [28]. Because AUR treatment significantly induced the expression of antioxidant enzyme transcripts, we investigated whether AUR prevents rotenone-induced ROS production in dopaminergic neuronal cells using the fluorescent dye DCFDA, which detects cytosolic ROS. We observed a 21.6% decrease in ROS levels in rotenone-exposed cells pretreated with 1 μM AUR compared with cells treated with rotenone only (Figure 3A,B), as assessed by flow cytometry. We then examined whether AUR treatment altered rotenone-induced mitochondrial superoxide production in SN4741 cells by adding the red fluorescent dye MitoSOX™ (which specifically targets mitochondrial superoxide) to rotenone- and AUR-treated cells, and quantified the results using flow cytometry. As shown in Figure 3C,D, mitochondrial superoxide levels in cells treated with rotenone only were comparable to those in AUR pretreated cells. These results are consistent with qPCR analyses, which showed that AUR specifically increased the transcription factor NRF2 and expression of its downstream targets, including Nqo1 and Gpx1, without affecting mitochondrial ROS scavenging enzymes, such as Sod1 and Sod2 (Appendix A). We found that AUR differentially regulates Gclc expression in the presence of rotenone or MPP^+^. We pretreated AUR for 1 h before treatment of neurotoxins to induce antioxidant enzyme expression. Although both rotenone and MPP^+^ targets complex I, rotenone showed higher inhibitory effect on mitochondrial respiration of SN4741 cells than MPP^+^, causing more ROS generation than MPP^+^. Increased ROS could offset against Gclc induction in rotenone treated cells. These results suggest that AUR induces expression of antioxidant enzymes, which act to effectively remove cellular ROS in dopaminergic neurons in the presence of neurotoxins, without altering mitochondrial ROS.

### 2.4. AUR Protects Neurotoxin-Induced Loss of Tyrosine Hydroxylase Expression

Tyrosine hydroxylase (TH) expression in the SN and projections of TH neurons to the striatum is reduced in association with progression of PD [29]. It has also been shown that MPTP-induced PD animal models show a loss of TH-positive neurons [30]. Accordingly, we determined whether AUR treatment protects against the loss of TH expression in the SN and striatum of MPTP-induced PD mice. AUR (25 mg/kg) or DMSO (vehicle control) was intraperitoneally injected into B6 mice 1 day before MPTP treatment (20 mg/kg, four times a day), and was then injected for two additional days. Using a brain slice preparation, we found a significant decrease in TH immunoreactivity in both the SN and striatum of mice injected with MPTP for 7 days compared with saline-injected mice. In contrast, TH immunoreactivity was preserved in AUR-pretreated mice (Figure 4A–D). Specifically, the number of TH-positive neurons was decreased by 43.4% in MPTP-injected mice compared with saline-injected mice, and was increased by 32% in AUR-treated mice compared with DMSO injected mice (Figure 4D).

It is known that AUR significantly decreases inflammation in the SN region of LPS-injected mice [14]. Because the number of reactive astrocytes in the SN is increased in MPTP-induced PD model mice [31], we examined whether AUR alleviates astrogliosis by immunofluorescence staining for the astrocyte marker, glial fibrillary acidic protein (GFAP). Because it is clear to show the neuroinflammation with astrocyte activation in this model as we previously reported [32], we chose the GFAP as a maker of neuroinflammation by MPTP. Whereas the relative GFAP intensity in the MPTP-only group was 3.3-fold higher than that in control mice, it was only 2.8-fold higher in the AUR-treated group, indicating a decrease in the number of reactive astrocytes (Figure 4E). These results suggest that AUR protects against the MPTP-induced reduction in TH expression and astrocyte activation.

### 2.5. AUR Ameliorates MPTP-Induced Motor Deficits

The nigrostriatal dopamine pathway is responsible for motor control, and TH activity is necessary for the release of dopamine, which regulates movement [33,34]. Because we found that AUR induces TH expression, we investigated the effect of AUR on movement deficits in MPTP-induced PD mice (Figure 5A). AUR-treated mice showed improved movement after MPTP injection compared with DMSO-treated mice, determined by monitoring behavior for 1 h in an open-field test (Figure 5B). Specifically, the total distance moved was decreased by 20.6% in MPTP-injected mice after 5 days compared with saline-injected mice (Figure 5C), whereas AUR-treated mice showed a 15.3% increase in movement distance compared with DMSO-treated mice (Figure 5C). Results presented in heat map form showed that AUR treatment significantly reduced residence time in the corner of the arena compared with that observed in mice treated with MPTP only (Appendix A). To further assess motor dysfunction, we performed vertical-grid tests of MPTP-injected and AUR-treated mice, as described by Kim et al. [35]. As shown in Figure 5D,E, MPTP-injected mice required 20 s longer to turn and a total of 25 s more time than control mice to complete the task. The time required to climb down was decreased by 5 s in MPTP-injected mice because of a 2-fold increase in missed steps compared with the control mice (Figure 5F,G). We found that AUR injection had no effect on the time to turn or total time, but restored the time to climb down to normal levels by decreasing missed steps observed in MPTP-only mice by 7% (Figure 5F,G). These findings suggest that AUR improves grip strength reduced by MPTP treatment.

Taken together, these results suggest that AUR mitigates motor dysfunction in MPTP-induced PD mice. As shown in Figure 6, we propose that AUR attenuates the effect of PD-related toxins on dopaminergic neurons through induction of NRF2 and expression of its target genes encoding antioxidant enzymes. AUR also increases mitochondrial respiration, which is suppressed in the presence of PD-related toxins (Figure 6). These protective effects of AUR on dopaminergic neurons consequently improve neurotoxin-induced motor deficits through preservation of TH expression.

## 3. Discussion

The complexity of PD and the variety of causative factors that contribute to its development create difficulties in identifying specific targets for effective treatments that might achieve complete disease remission. In the present study, we focused on modulation of mitochondrial energy metabolism and inhibition of ROS production by damaged mitochondria using the natural compound AUR. We postulate a dual preventive mechanism of AUR: (1) Induction of expression of genes encoding antioxidant enzymes, which protect against ROS, and (2) reduction of mitochondrial respiration by neurotoxins. 

The lack of available treatment options for preventing or slowing the progression of PD has driven increased efforts to delay the occurrence of PD symptoms—the primary concept in current drug development strategies [36]. One disease-modifying agent, vitamin E, counteracts oxidative stress, and its intake is inversely correlated with PD occurrence [37]. In addition, the green tea polyphenol, (–)-epigallocatechine-3-gallate [38], and two Mediterranean plant-based extracts, *Padina pavonica* (EPP) and *Opuntia ficus-indica* (EOFI), ameliorate neurodegeneration in PD [39]. However, the mechanisms by which these treatments affect PD pathogenesis have not been identified. Unlike these latter studies, we focused specifically on mitochondrial respiration—considered the first target of environmental causative factors such as paraquat—and ROS overproduction by damaged mitochondria [36]. We assessed the protective effect of AUR by measuring mitochondrial oxygen consumption rate (OCR) and antioxidant enzyme expression levels in a neuronal cell line model of mitochondrial toxicity. We found that the overall changes in cellular metabolism induced by AUR are just a slight change in mitochondrial respiration. In the AUR-pretreated and MPP^+^-treated groups, basal OCR was higher than that of the control. However, there was no significant difference in behavioral tests such as the open-field test and the vertical grid test between control and AUR-treated groups (Figure 5). These results suggest that AUR increases OCR of dopaminergic neurons in the presence of MPP^+^ and it is consequently sufficient to improve MPTP-induced PD-like behavior to a normal level. But, additive beneficial effects on behavior or hypermobility were not found. Therefore, AUR could be used for prevention purposes by reducing adverse effects. Thus, our findings suggest that AUR, a coumarin from a source as simple and natural as citrus peel oil, could assist in preventing PD.

In general, enhancing mitochondrial respiration is expected to increase ROS generation, because the mitochondrial respiratory chain is a major source of intracellular ROS production and many enzymes that convert molecular oxygen to ROS are present in mitochondria [40]. Impairment of mitochondrial respiration plays a major role in the pathogenesis of PD, and increased ROS levels are known to be among the important causes of PD [40]. The key strength of AUR is its dual function described above, which enables AUR to protect a decrease in mitochondrial respiration caused by neurotoxins without increasing cellular ROS, although how these two effects are linked is not yet clear.

In a previous study, we reported that AUR suppresses mitochondrial respiration in the renal cell carcinoma cell line, RCC4 [11]. It has also been reported that AUR acts as a mitochondrial poison in the T-47D human breast cancer cell line [8]. However, our study suggests that AUR increases mitochondrial function in PD-like conditions. Although these two observations are seemingly at odds, they might actually be compatible, given that cancer cells possess exceptional cellular pathways compared with normal cells. Activation of NRF2 has been reported in several types of cancer cells [41]. NRF2, which is responsive to oxidative stress, is constitutively expressed in normal cells, but its protein level is low because of KEAP1-mediated ubiquitination and degradation [42]. Considering that AUR acts, at least in part, through induction of NRF2, its actions on cellular pathways could be different in cancer cells and normal cells. It is also worth noting that the AUR concentration range was significantly different between these two studies. In the cancer cell study, cellular metabolism was targeted by inhibiting translation of the HIF-1α transcription factor using an AUR concentration of 100 μM. At a high concentration, AUR reduced basal OCR to 67% of that in untreated cancer cells, which show immature mitochondrial function. In the current study, we tested AUR at a concentration of 1 μM, and found that it increased basal OCR in dopaminergic neuron-like cells in the presence of neurotoxins. Notable in this context, some antioxidants, including EGCG, have been reported to show neuroprotective activity at low concentrations, but pro-oxidant activity at high concentrations [38]. 

We also suggest the potential of AUR in trials of combined therapy with levodopa. Levodopa is one of the main drugs used for relief of PD symptoms, but it should be used with caution in younger patients with early PD [36,43]. If there were a drug that could prevent progression of the disease, it should be used starting as early as possible. Although drugs currently used in combination with levodopa, such as benserazide and carbidopa, reduce the peripheral effects of levodopa and increase levodopa concentrations in the brain [36], combination therapy with AUR would provide additional neuronal protective effects through a different pathway. If an early diagnosis of pre-symptomatic PD patients is possible in the near future, AUR could be beneficial to delay the loss of dopaminergic neurons and PD-behavior symptoms. Combining these drugs in a single therapeutic regimen would seek to relieve symptoms while delaying disease progression. 

## 4. Materials and Methods

### 4.1. Cell Culture

SN4741 mouse embryonic substantial nigra dopaminergic neuronal cell line was cultured in RF media containing Dulbecco’s modified Eagle’s medium (DMEM, Welgene, Korea), 10% FBS (Hyclone, MA, USA), 1% penicillin and streptomycin (Hyclone, MA, USA), 0.6% D-glucose and 0.7% 200 mM·L-glutamine at 33 °C under 5% CO_2_ and 21% O_2_ condition. 

### 4.2. Measurement of Cell Viability

In the sulforhodamine B assay, SN4741 cells (5 × 10^3^ cells per well) were seeded in triplicate in 96-well plates and incubated overnight. Added to each well were media containing Rot (0, 0.5, 1 and 10 uM, Sigma-Aldrich, MO, USA) for 6 h or MPP^+^ (0, 1, 4, 8 mM, Sigma-Aldrich, MO, USA) for 24 h in the presence or absence of AUR 1 uM (Sigma-Aldrich, MO, USA). The media were removed and cells were fixed with 10% TSA at 4 °C for 1 h. After washing, the cells were incubated with 0.4% SRB (Sigma-Aldrich, MO, USA) solution at room temperature for 20 min. The wells were washed with 1% acetic acid five times and dried in air. After resolving the proteins with 10 mM unbuffered Tris, absorbance was read at 490 nm using a Multiskan Ascent plate reader.

### 4.3. Flow Cytometry

For analyzing ROS generation, the fluorescent dye, MitoSOX™ red reagent (Invitrogen, CA, USA) and DCFDA (Invitrogen, CA, USA) were used following the manufacturers’ instructions. SN4741 cells (2–4 × 10^5^ cells in 60 mm dish) were incubated with Rot for 6 h and AUR was pretreated for 1 h. Media was discarded and washed with HBSS and incubated for 30 min in the dark with DCFDA or MitoSOX™ (5 μM final concentration). Cells were washed with PBS and trypsinized, then resuspended in PBS/EDTA. After washed with PBS, cells were collected and kept on ice in the dark for immediate detection with the flow cytometer. Fluorescence was measured on a FACScan (BD Biosciences, NJ, USA) using excitation/emission wavelengths of 485/535 nm, and 510/580 nm for DCFDA and MitoSOX™, respectively. The values were expressed as mean fluorescence of the cell population.

### 4.4. Measurement of Oxygen Consumption Rate (OCR)

SN4741 cells cultured with rotenone or MPP^+^ ± treatment with AUR 2 uM were plated 2 × 10^4^ cells at each well. Basal OCR was analyzed by XF24 analyzer (Seahorse, MA, USA). Then, 20 µg/mL of oligomycin A (an ATPase inhibitor, Sigma-Aldrich, MO, USA), 50 µM of carbonyl cyanide 3-chlorophenylhydrazone (CCCP, an uncoupler, Sigma-Aldrich, MO, USA) and 20 µM rotenone (a mitochondrial complex I inhibitor, Sigma-Aldrich, MO, USA) were sequentially added into each well and OCR was measured at 37 °C.

### 4.5. RNA Isolation and Real Time PCR

Total RNA was isolated using Trizol from SN4741 cells treated with Rot (0, 0.5 or 1 uM) or MPP^+^ (0, 50, 75 or 100 uM) and AUR for 24 h. cDNA was synthesized from total RNA with 5× RT premix. After mixing cDNA, primers and SYBR mix, mRNA expression was analyzed using a Rotor Gene 6000 system (Corbett Life Science, Venlo, Netherlands) and normalized to 18s rRNA. Primers used in this study: NRF2, 5′-CCAGAAGCCACACTGACAGA-3′ (forward) and 5′-GGAGAGGATGCTGCTGAAAG-3′ (reverse); NQO1, 5′-TTCTCTGGCCGATTCAGAGT-3′ (forward) and 5′-GGCTGCTTGGAGCAAAATAG-3′ (reverse); GPX, 5′- GTCCACCGTGTATGCCTTCT-3′ (forward) and 5′-TCTGCAGATCGTTCATCTCG-3′ (reverse); GST, 5′-GGCATCTGAAGCCTTTTGAG-3′ (forward) and 5′-GAGCCACATAGGCAGAGAGC-3′ (reverse); Gclc, 5′-AGGCTCTCTGCACCATCACT-3′ (forward) and 5′- TGGCACATTGATGACAACCT-3′ (reverse); Gclm, 5′-TGGAGCAGCTGTATCAGTGG -3′ (forward) and 5′-AGAGCAGTTCTTTCGGGTCA-3′ (reverse); GR, 5′-CACGACCATGATTCCAGATG-3′ (forward) and 5′-CAGCATAGACGCCTTTGACA-3′ (reverse); 18s rRNA, 5′-CGACCAAAGGAACCATAACT-3′ (forward) and 5′-CTGGTTGATCCTGCCAGTAG-3′ (reverse).

### 4.6. Animal Experiments

Temperature was maintained to 22 °C and light condition was adjusted to a 12 h light-dark cycle. Animal experiments were approved by the Institutional Animal Care and Use Committee of Chungnam National University. The ethical approval number is CNU-00912 and approval date is March-1-2017. To establish the MPTP-induced PD mouse model, C57BL/6 mice (8-week-old, male) were intraperitoneally injected with MPTP (1-methyl-4phenyl-1.2.3.6-tetrahydropyridine, Sigma-Aldrich, MO, USA, 2 mg/mL in saline, 20 mg/kg for one injection) four times with 2 h intervals in a day. Control mice were injected with saline. Before 24 h and 48 h of MPTP injection, auraptene (25 mg/kg) was injected intraperitoneally.

### 4.7. Immunofluorescence Staining and Immunohistochemistry

Saline and MPTP injected Mice were perfused and fixed with 4% paraformaldehyde (PFA). The whole brain was dipped in the 4% PFA and then moved to 30% sucrose solution to dehydrate for three days. The samples were frozen and sectioned, 25 μm of each slice. For the immunofluorescence staining, after 15 min of PBS washing, sections were blocked for 1.5 h with 3% donkey serum (Dako, Glostrup, Denmark) and 0.3% triton x-100 with PBS. Then, sections were incubated with anti-TH antibody (Millipore, MA, USA), anti-GFAP (1:1000, Abcam, Cambridge, UK) diluted with blocking solution overnight at 4 °C. Sections were washed with PBS and incubated with anti-mouse Alexa 594 and anti-chicken Alexa 488-conjugated anti-IgG secondary antibodies containing solution for 1 h at room temperature. For immunohistochemistry, brain slices were incubated with anti-TH antibody for overnight at 4 °C and then incubated with a secondary antibody (Dako EnVision^+^ system-HRP, CA, USA) for 1 h. The slices were reacted with DAB^+^ substrate buffer. After mounting with mounting medium (Dako North America Inc., CA, USA), the slides were visualized using an IX70 confocal microscope (Olympus, Tokyo, Japan).

### 4.8. Protein Isolation and Western Blotting 

The protein of mice tissues and SN4741 cells, treated with Rot (0, 0.5 or 1 uM) or MPP^+^ (0, 50, 75 or 100 uM) and pretreated with 10 uM Auraptene or DMSO for 1 h, were extracted using RIPA buffer (1% Nonidet P-40, 0.1% SDS, 150 mM NaCl, 50 mM Tris–HCl pH 7.5 and 0.5% deoxycholate) with 10% of phosphatase inhibitor and protease inhibitor (Roche, Basel, Switzerland). Equal amounts of proteins were loaded on SDS-PAGE gel and run by electrophoresis. After, they were transferred to polyvinylidene fluoride (PVDF) membrane, blocked by 5% skim milk for 1 h. Then, membranes were incubated with primary antibody including anti-NRF2 (Santa Cruz Biotechnology, CA, USA) and anti-α-Tubulin (Santa Cruz Biotechnology, CA, USA) antibody at 4 °C overnight. Anti-IgG horseradish peroxidase antibody (Pierce Biotechnology, MA, USA) correspond with the host of primary antibody was used as secondary antibody. Protein bands were detected by ECL system (Thermo Scientific, MA, USA). 

### 4.9. Behavior Test

Open-field test: Mice were placed in a 40 × 40 × 40-box respectively. Movement was recorded for 1 h and analyzed with EthoVision XT 11.5 software.

Vertical grid test: The vertical grid test was performed following the previous study [35]. For performing the vertical grid test, mice were habituated to the apparatus. After habituation for 3 days, a mouse was placed inside the apparatus and was allowed to turn and climb down. The movement was recorded.

### 4.10. Statistical Analysis

All data are represented as mean values ± SEM (error bars). The statistical analysis of data was performed using Prizm version 5 software (Graphpad, CA, USA). Significance of differences between two groups were analyzed by one-tailed student’s t-test. A P value <0.05 was considered statistically significant.

## Figures and Tables

**Figure 1 ijms-20-03409-f001:**
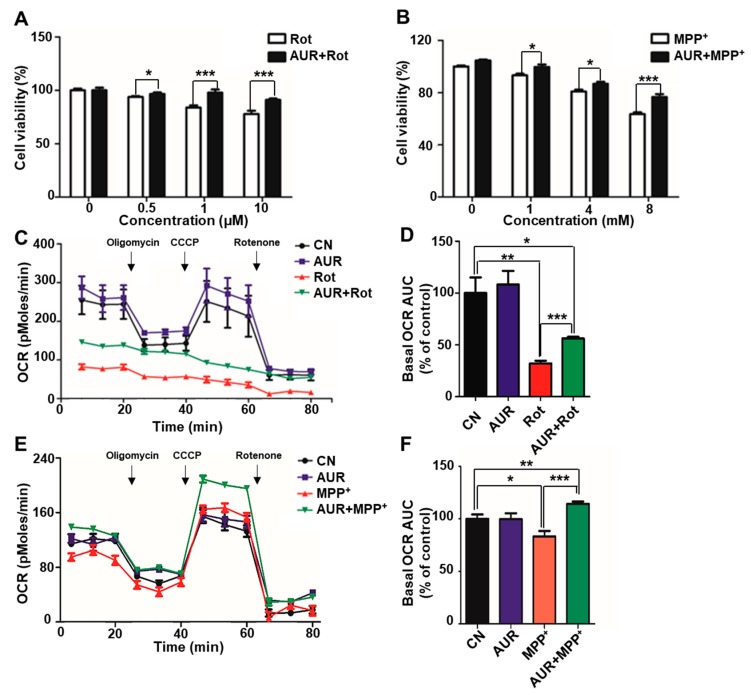
Auraptene (AUR) increases SN4741 cell viability and oxygen consumption rate (OCR) in the presence of neurotoxins. (**A**,**B**) SN4741 cells (5 × 10^3^) plated in 96-well plates were incubated in media containing different concentrations (0, 0.5, 1 or 10 μM) of rotenone (Rot) for 6 h or MPP^+^ (0, 1, 4, or 8 mM) for 24 h in the presence or absence of AUR (1 μM). Cell viability was measured by sulforhodamine B (SRB) assay after 6 or 24 h of drug treatment. (**C**–**F**), OCR was measured in SN4741 cells cultured with rotenone (**C**,**D**) or MPP^+^ (**E**,**F**), with or without treatment with AUR. (**D**,**F**) Basal OCR area under the curve was calculated using XF24 analyzer software. Values are presented as means ± SD (bars) of triplicate samples (* *P* < 0.05, ** *P* < 0.01, *** *P* < 0.001 vs. corresponding controls). CN, control.

**Figure 2 ijms-20-03409-f002:**
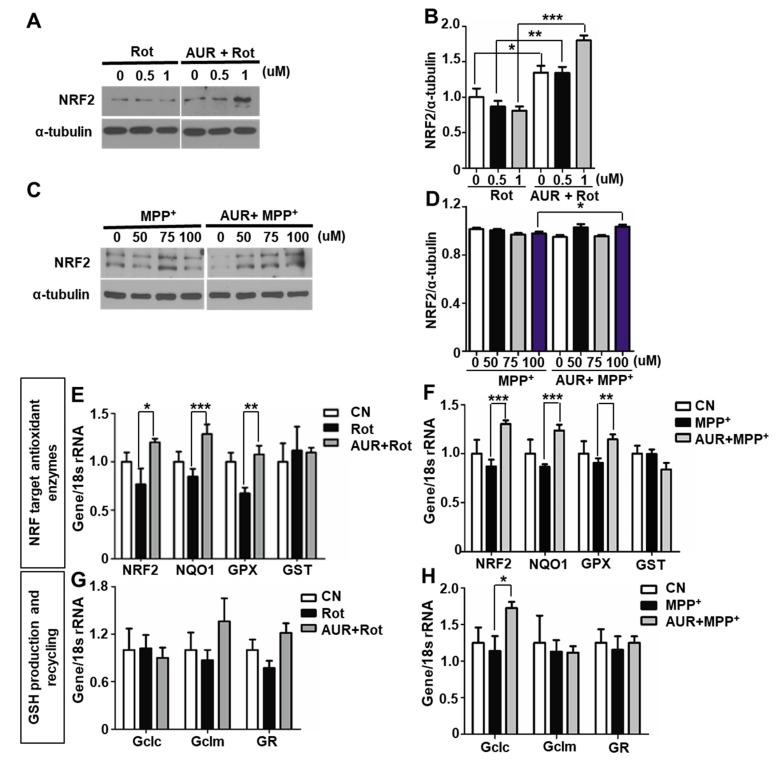
AUR induces expression of genes encoding antioxidant enzymes. (**A**–**D**) SN4741 cells were incubated in media containing different concentrations (0, 0.5 or 1 μM) of rotenone (Rot) or MPP^+^ (0, 50, 75 or 100 μM), with or without pretreatment for 1 h with 10 μM AUR or DMSO. NRF2 protein expression was determined by Western blotting after 24 h (**A**) or 6 h (**C**) of drug treatment. The band intensity of NRF2 was measured using the ImageJ program (**B**,**D**). (**E**–**H**) Expression of mRNA for NRF2 target antioxidant enzymes (**E**,**F**) and GSH recycling-related genes (**G**,**H**) were assessed after a 24 h drug treatment using qPCR. Values are presented as means ± SD (bars) of triplicate samples (* *P* < 0.05, ** *P* < 0.01, *** *P* < 0.001 vs. corresponding controls).

**Figure 3 ijms-20-03409-f003:**
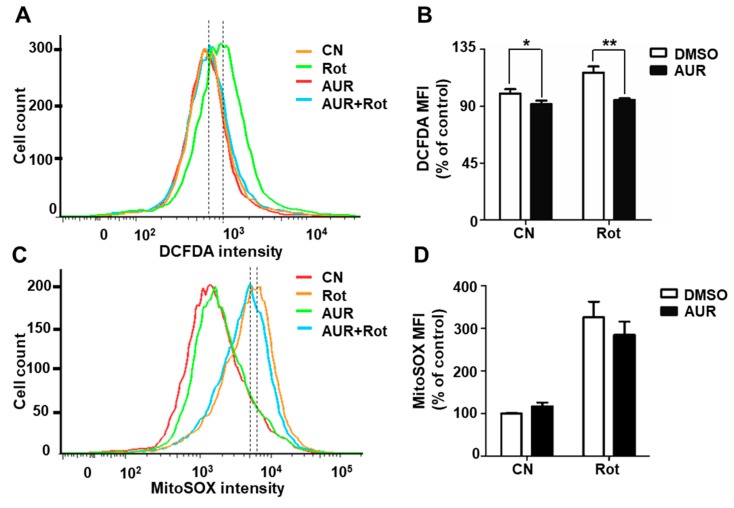
AUR protects against rotenone-induced ROS production. (**A**–**D**) SN4741 cells were incubated with rotenone (Rot) for 6 h, with or without AUR pretreatment for 1 h. Cells were stained with DCFDA or MitoSOX™, and fluorescence intensity was measured by flow cytometry. Total ROS was determined by measuring DCFDA-stained cells (**A**,**B**), and mitochondrial ROS was determined by measuring MitoSOX™-stained cells (**C**,**D**). Median fluorescence intensity (MIF) values are presented as means ± SD of three experiments (* *P* < 0.05, ** *P* < 0.01 vs. corresponding controls). CN, control.

**Figure 4 ijms-20-03409-f004:**
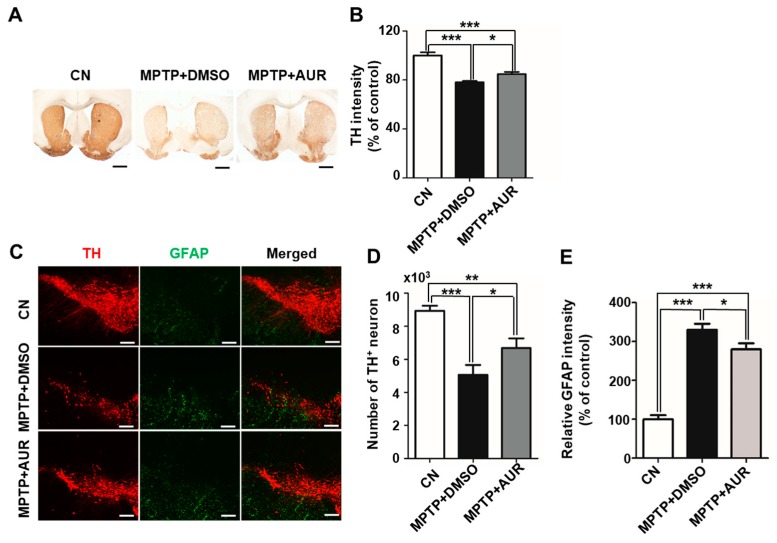
Pretreatment with AUR reduces MPTP-induced loss of TH expression in the SN and striatum. (**A**) Immunohistochemical detection of TH in the striatum of C57BL/6 mice injected with MPTP (20 mg/kg, i.p.) or saline, together with AUR (25 mg/kg, i.p.) or DMSO. Scale bars: 50 μm. (**B**) TH expression was decreased in MPTP-injected mice, an effect that was attenuated by AUR cotreatment. TH intensity was measured using ImageJ, and results are presented as a percentage of control values. (**C**) Immunofluorescence detection of TH in the SN region. TH-positive dopaminergic neurons (red) and astrocytes (green) were visualized by confocal microscopy. (**D**,**E**) Number of TH-positive neurons was calculated, and relative GFAP intensity was measured using ImageJ. Data are presented as means ± SD of three experiments (*n* = 10/group; * *P* < 0.05, ** *P* < 0.01, *** *P* < 0.001 vs. corresponding controls). CN, control. Scale bars: 500 μm.

**Figure 5 ijms-20-03409-f005:**
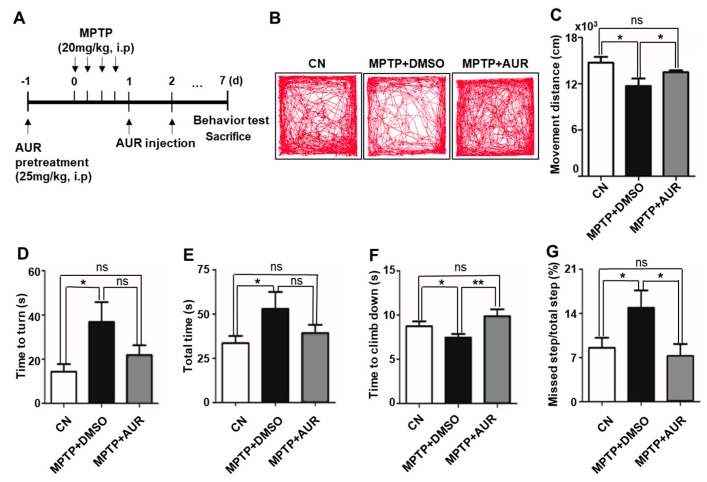
AUR improves MPTP-induced movement disorders. (**A**) Experimental timeline of AUR injection into the MPTP-induced mouse model of PD and behavioral tests. Mice were intraperitoneally injected with MPTP (20 mg/kg) 24 h after AUR (25 mg/kg) administration; AUR was further injected 24 h and 48 h after MPTP injection. Open-field and vertical-grid tests were performed after 7 days of MPTP injection. (**B**) Tracks visualizing mouse movements for 1 h are presented. Eight-week-old MPTP-induced PD mice showed a decrease in movement compared with control mice, whereas AUR-cotreated mice showed improveed movement (*n* = 5/group). (**C**) Total distance moved in 1 h was determined using EthoVision software and is presented as means ± SD. (**D**–**G**) Mice were placed at the bottom of the vertical grid and allowed to climb upward while movement was recorded. Time to turn (**D**), total climbing time (**E**), time to climb down (**F**), and percentage of total steps missed (**G**) were calculated. Values are presented as means ± SD (*n* = 5/group; * *P* < 0.05, ** *P* < 0.01 vs. corresponding controls; ns, not significant). CN, control.

**Figure 6 ijms-20-03409-f006:**
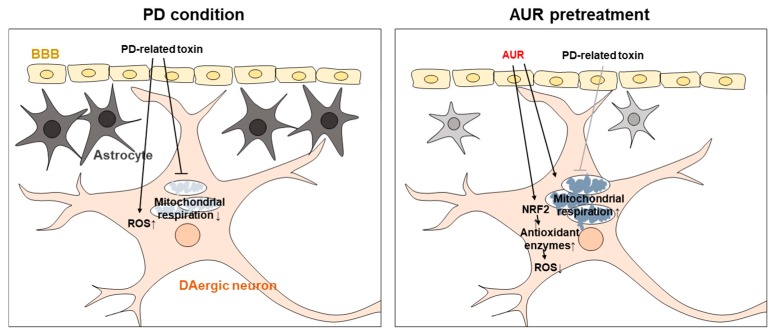
Schematic representation of the dopaminergic neuron-protective mechanism of AUR in a PD model. AUR alleviates neurotoxin-induced oxidative stress in dopaminergic neurons by stimulating the transcription factor NRF2 and inducing expression of downstream genes encoding antioxidant enzymes. Inhibition of mitochondrial respiration by PD-related toxins is mitigated by AUR treatment. AUR protects dopaminergic neurons against neurotoxins and ameliorates PD-like behavior.

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
