# Peer review of "Auraptene Mitigates Parkinson’s Disease-Like Behavior by Protecting Inhibition of Mitochondrial Respiration and Scavenging Reactive Oxygen Species"

_ijms, 2019, doi:10.3390/ijms20143409_

Round 1

Reviewer 1 Report

The manuscript of Jang et al. describes the protective effects of auraptene in mitigating the toxic effects of the toxins used to induces Parkinson’s disease. The strength of this work is the design of the experiments that includes both cell cultures and animal models.

Nevertheless, the two cell lines, the animal models, the two toxin causes some confusion to the reader especially because part of the Material and Method and the results are included in Supplementary material, avoiding a clear evaluation of the entire work. For these reasons, there are a few issues I would like to see addressed.

Because of the use of two cell lines and animal model and two toxins, I suggest adding a paragraph in Materials and Methods or at the end of the introduction to briefly summarize the purpose of the study. Sincerely, it was difficult to understand the work as a single process.

Could the authors explain the purpose of the use of two toxins to induce PD? Both the models possess advantages and disadvantages, but the reasons for their use are not clear in the manuscript.

Page 2 Line 4: recently it was published a paper analyzing early pathological mechanisms (https://doi.org/10.3390/ijms20092224), although the mitochondrial toxicity was not analyzed (rotenone, used to develop the pathology, is a well-known mitochondrial toxin).

Pages 6 - 7: is it more correct to say that AUR induces TH expression or that AUR reduce the cell death induced by MPTP? To be sure about its activity as inducer it should be applied after the induced damages to observe induction of TH expression. These results show a reduced loss of TH-positive cells probably due to induction of mechanisms of resistance against the toxin.

Materials and Methods: Could the authors explain why rotenone was applied to the cells only for 6 hours for cell viability while MPP for 24 hours?

Page 7: the analysis of microglia activation is also a great marker of neuroinflammation. Why did the authors choose only GFAP?

Materials and Methods: to clarify the protocol, the PD mouse model was induced by four injections of MPTP, this means one day of treatment. The model is an acute model of MPTP, while PD in human is chronically developed in many years. Why the authors choose this protocol instead of toxicity induced in more days?

Page 10: the hypothesis that AUR increases mitochondrial functions seems to not be a conclusion of the results of the manuscript. In fact, except for a peak in 40-60 minutes in Figure 1 E, AUR seems to normalize the value of mitochondrial impairment induced by the toxins, while when applied alone it shows no relevant effects.

Page 10: only a precision about the final discussion and the suggestion about the co-treatment: when the first PD symptoms are recognized and Levodopa treatment starts, at least 40-60% of the dopaminergic neurons are lost. This means that a neuroprotective drug like AUR or antioxidant/anti-inflammatory treatments should start before the loss of cells causes the symptomatic phase. For this reason, investments in an early diagnosis are indispensable to recognize the first signs of damages during the asymptomatic phase.

Figure 1 A-B: it should be better to specify in the x-axis that "Concentration" is "Concentration Rot".

Figure 1D: Pay attention to the unit in y-axis: 100 instead of 10. In addition, could the authors explain where AUR induces a 30% increases in OCR when the changes observed in graph D and E are far from this value?

Supplementary Figure 3 A: No complete legend present.

Author Response

6th July, 2019

International Journal of Molecular Sciences Editorial Office

Dear Dr. Owen Chen

We would like to thank the reviewers for the detailed comments on our manuscript (ijms-539563). Please find enclosed our revised manuscript, entitled “Auraptene mitigates Parkinson’s disease-like behavior by enhancing mitochondrial respiration and scavenging reactive oxygen species” which we would like to submit for publication in International Journal of Molecular Sciences.

We are grateful to the reviewers and editors for their suggestions. We believe that the changes made in response to these suggestions have strengthened our manuscript, including revising our findings and their interpretation. We have revised our manuscript to answer the point-by-point response to all comments raised by the reviewers. The changed manuscript is noted by blue colored text.

Sincerely yours,

Please address all correspondence to:

Gi Ryang Kweon, M.D, Ph.D

Professor

Department of Biochemistry,

College of Medicine, Chungnam National University

Munhwa-dong, Jungu, Deajeon 301-747, Republic of Korea

Phone: 82-42-580-8226

Fax: 82-42-580-8121

E-mail: mitochondria@cnu.ac.kr

Jun Young Heo, M.D, Ph.D

Associate Professor

Department of Biochemistry,

College of Medicine, Chungnam National University

Munhwa-dong, Jungu, Deajeon 301-747, Korea

Phone: 82-42-580-8351

Fax: 82-42-580-8121

E-mail: junyoung3@gmail.com

Responses are in bold

Reviewer 1

The manuscript of Jang et al. describes the protective effects of auraptene in mitigating the toxic effects of the toxins used to induces Parkinson’s disease. The strength of this work is the design of the experiments that includes both cell cultures and animal models.

Nevertheless, the two cell lines, the animal models, the two toxin causes some confusion to the reader especially because part of the Material and Method and the results are included in Supplementary material, avoiding a clear evaluation of the entire work. For these reasons, there are a few issues I would like to see addressed.

 We appreciated the reviewers for the constructive comments. We have addressed each concern of the reviewers in the appended rebuttal document.

 We thank to the reviewer’s suggestion. We realized the Supplementary figure 4, the experiment with SH-SY5Y causes some confusion and is beside the point in our manuscript. Although it shows the increase of NRF2 expression in SH-SY5Y-treated AUR+Rot compared to Rot group, it is better to remove the figure to help to understand our manuscript without confusion.

Because of the use of two cell lines and animal model and two toxins, I suggest adding a paragraph in Materials and Methods or at the end of the introduction to briefly summarize the purpose of the study. Sincerely, it was difficult to understand the work as a single process.

 We appreciated the suggestion. To clarify the purpose, we revised the introduction to the following sentences. Page 5, line 5 “In order to clarify the antioxidative effect by treatment of AUR, we investigated the alteration of cell viability, antioxidant enzyme expression and ROS generation by using rotenone, MPP+ in SN4741 cell line. And we demonstrated that pretreatment with AUR improves movement deficits in association with an increase in the number of dopamine neurons in the substantia nigra (SN) of MPTP-induced PD mouse model which inhibit the mitochondrial complex I. On the basis of these findings, we suggest that AUR pretreatment acts through protection of decrease in mitochondrial respiration by neurotoxin and down-regulation of ROS of dopaminergic neuron to produce its beneficial PD-related neurobiological changes.

Could the authors explain the purpose of the use of two toxins to induce PD? Both the models possess advantages and disadvantages, but the reasons for their use are not clear in the manuscript.

 Rotenone, a mitochondrial complex I inhibitor produces ROS and induces oxidative stress. After MPTP is converted to MPP+ in glial cells, it causes ROS and mitochondrial dysfunction in dopaminergic neurons. It is suitable to show the effects of AUR in PD model which results from mitochondrial dysfunction because both of two toxins lead to PD by inducing oxidative stress. Besides MPTP-induced PD mouse model is established very well [1], also we have used the model in the previous paper[2]. We added this explanation to introduction section paragraph 2. Page 4, line 16. “These two neurotoxins are suitable to show the effects of AUR in PD model which results from mitochondrial dysfunction because both of two toxins lead to PD by inducing oxidative stress.”

Page 2 Line 4: recently it was published a paper analyzing early pathological mechanisms (https://doi.org/10.3390/ijms20092224), although the mitochondrial toxicity was not analyzed (rotenone, used to develop the pathology, is a well-known mitochondrial toxin).

Thank you for your suggestion. We added the reference in the following part of introduction. Page 4, line 4. ‘To overcome the limitation of PD drugs, arising focus on PD studies are early pathological changes which regard to establishment of a framework for the preclinical and prodromal stages of PD [3]

Pages 6 - 7: is it more correct to say that AUR induces TH expression or that AUR reduce the cell death induced by MPTP? To be sure about its activity as inducer it should be applied after the induced damages to observe induction of TH expression. These results show a reduced loss of TH-positive cells probably due to induction of mechanisms of resistance against the toxin.

We thank the reviewer for pointing this out. As we mentioned on introduction, the AUR inhibits dopaminergic neuronal loss in the lipopolysaccharide mouse model [4]. Our results show AUR reduced the loss of TH-positive cells through the pre-treatment of AUR not only post-treatment. It is reported that increase of ROS induces dopaminergic neuronal cell death due to oxidative stress in PD patient SN and striatum[5]. As far as we tested, AUR seems to protect dopaminergic neuronal loss by reducing neurotoxin- induced ROS generation. We do not know the direct effect of induction of TH by AUR treatment, but enhancement of antioxidant capacity and mitochondrial resipiration are more subject to resistance against toxin.

Materials and Methods: Could the authors explain why rotenone was applied to the cells only for 6 hours for cell viability while MPP for 24 hours?

Protective effect of AUR on the neurotoxin- induced decrease of cell viability is observed until 12 hours of incubation. However, rotenone inhibits mitochondrial complex I with higher efficiency of MPP+ as shown in mitochondrial respiration measured by OCR (Fig. 1). Therefore, we tested AUR effect after 6 hours of rotenone treatment and 24 hours of MPP+ treatment, respectively.

Page 7: the analysis of microglia activation is also a great marker of neuroinflammation. Why did the authors choose only GFAP?

We appreciated the comment. We have confirmed the microglia activation with GFAP activation in MPTP-induced PD model in our previous study [2]. Because it is more clear to show the neuroinflammation in this model, we chose the GFAP as a maker of neuroinflammation. We added this explanation and reference to result section on page 9.

Materials and Methods: to clarify the protocol, the PD mouse model was induced by four injections of MPTP, this means one day of treatment. The model is an acute model of MPTP, while PD in human is chronically developed in many years. Why the authors choose this protocol instead of toxicity induced in more days?

Thank you for the comment. As we mentioned before, the MPTP-induced PD model is established well in research of PD [1]. This model shows the PD-like symptoms such as a decrease of TH expression caused mitochondrial dysfunction in dopaminergic neurons. Although there are chronic MPTP-induced PD model, the model shows protein and lipid inclusions that are alpha-synuclein positive [6]. That’s why we chose the acute MPTP-induced PD model to focus on AUR protective effect for oxidative stress and mitochondrial dysfunction.

Page 10: the hypothesis that AUR increases mitochondrial functions seems to not be a conclusion of the results of the manuscript. In fact, except for a peak in 40-60 minutes in Figure 1 E, AUR seems to normalize the value of mitochondrial impairment induced by the toxins, while when applied alone it shows no relevant effects.

We thank for the comment. We agree with the reviewer’s opinion that AUR effect on the mitochondrial respiration seems protection of reduction by neurotoxin. Therefore, we edited description in abstract, result and discussion section.

Page 10: only a precision about the final discussion and the suggestion about the co-treatment: when the first PD symptoms are recognized and Levodopa treatment starts, at least 40-60% of the dopaminergic neurons are lost. This means that a neuroprotective drug like AUR or antioxidant/anti-inflammatory treatments should start before the loss of cells causes the symptomatic phase. For this reason, investments in an early diagnosis are indispensable to recognize the first signs of damages during the asymptomatic phase.

We agree with your opinion and revised the part of discussion to highlight the effectiveness of AUR co-treatment to pre-symptomatic phase patients as you mentioned. Page 11, line 30. “If an early diagnosis of pre-symptomatic PD patients is possible in the near future, AUR could be beneficial to delay the loss of dopaminergic neurons and PD-behavior symptoms.”

Figure 1 A-B: it should be better to specify in the x-axis that "Concentration" is "Concentration Rot".

 As the reviewer’s comment, we revised the labeling in figure 1 A-B.

Figure 1D: Pay attention to the unit in y-axis: 100 instead of 10. In addition, could the authors explain where AUR induces a 30% increases in OCR when the changes observed in graph D and E are far from this value?

We thank to the reviewer’s comment. We edited y-axis labeling of Fig. 1D. Area under curve (AUC) is automatically calculated by XF24 software. AUC of basal OCR in AUR pre-treated and rotenone treated group showed 24.1 % higher than rotenone only treated group as shown in Fig. 1D. And, AUC of basal OCR in AUR pre-treated and MPP+ treated group showed 31.03 % higher than MPP+ only treated group as shown in Fig. 1E-F.

Supplementary Figure 3 A: No complete legend present.

We thank to the reviewer’s comment. We edited the legend to clarify the description of data.

References

1.        Jackson-Lewis, V.; Przedborski, S., Protocol for the MPTP mouse model of Parkinson's disease. Nature protocols 2007, 2, (1), 141-51.

2.        Kim, S. J.; Ryu, M. J.; Han, J.; Jang, Y.; Lee, M. J.; Ju, X.; Ryu, I.; Lee, Y. L.; Oh, E.; Chung, W.; Heo, J. Y.; Kweon, G. R., Non-cell autonomous modulation of tyrosine hydroxylase by HMGB1 released from astrocytes in an acute MPTP-induced Parkinsonian mouse model. Laboratory Investigation 2019.

3.        Dal Ben, M.; Bongiovanni, R.; Tuniz, S.; Fioriti, E.; Tiribelli, C.; Moretti, R.; Gazzin, S., Earliest Mechanisms of Dopaminergic Neurons Sufferance in a Novel Slow Progressing Ex Vivo Model of Parkinson Disease in Rat Organotypic Cultures of Substantia Nigra. International Journal of Molecular Sciences 2019, 20, (9), 2224.

4.        Okuyama, S.; Semba, T.; Toyoda, N.; Epifano, F.; Genovese, S.; Fiorito, S.; Taddeo, V. A.; Sawamoto, A.; Nakajima, M.; Furukawa, Y., Auraptene and Other Prenyloxyphenylpropanoids Suppress Microglial Activation and Dopaminergic Neuronal Cell Death in a Lipopolysaccharide-Induced Model of Parkinson's Disease. Int J Mol Sci 2016, 17, (10).

5.        Dias, V.; Junn, E.; Mouradian, M. M., The role of oxidative stress in Parkinson's disease. J Parkinsons Dis 2013, 3, (4), 461-91.

6.        Meredith, G. E.; Rademacher, D. J., MPTP mouse models of Parkinson's disease: an update. J Parkinsons Dis 2011, 1, (1), 19-33.

7.        Claudia, T.; Christine, C. I. I.; A., T. D., Transcriptional Regulation by Nrf2. Antioxidants & Redox Signaling 2018, 29, (17), 1727-1745.

8.        Prince, M.; Li, Y.; Childers, A.; Itoh, K.; Yamamoto, M.; Kleiner, H. E., Comparison of citrus coumarins on carcinogen-detoxifying enzymes in Nrf2 knockout mice. Toxicol Lett 2009, 185, (3), 180-6.

9.        Jang, Y.; Han, J.; Kim, S. J.; Kim, J.; Lee, M. J.; Jeong, S.; Ryu, M. J.; Seo, K. S.; Choi, S. Y.; Shong, M.; Lim, K.; Heo, J. Y.; Kweon, G. R., Suppression of mitochondrial respiration with auraptene inhibits the progression of renal cell carcinoma: involvement of HIF-1alpha degradation. Oncotarget 2015, 6, (35), 38127-38.

Reviewer 2 Report

The article is quite well done, the experiments and outcomes are well described and clearly organized in the paper. However, some minor revisions are needed. After repairing these missing details and inaccuracies, the manuscript can be accepted for publication.

Page 3, figure 1

-        What does CN stands for, the control? Please specify.

If CN is indeed the control: figure 1E/1F are % of control, why is the CN bar in 1F not at 100%?

-        Triplicate samples: does it mean the measurements are performed in triple technical replicates? Or was the experiment repeated three times (independently, with new cells)? Please make clear both how many repetitions and how many replicates were used.

-        Figure 1D: the Y-axis labels should probably be 150/100/50 (instead of 150/10/50).

-        The neuroprotective effect of AUR seems to be stronger after Rot treatment, compared to after MPP+ treatment (according to OCR results, …). Can you give an explanation why this could be the case?

-        Figure 1F: the AUR-treated groups show a higher bacal OCR than the control group,  is it significantly higher and does it have any consequences? Can you explain that?

Page 4

-        NRF2 is mentioned to be involved in regulation of antioxidant-related genes, how? Does it activate/suppress?

Similarly: “These results indicate that AUR treatment induces NRF2 protein expression in cells.” But what is the consequences of this increased protein expression?

-        Here the SH-SY5Y is mentioned for the first time. Why those cells, how do they relate/are they different from the SN4741 cells?

In general: the motivation for the choice of those two cell lines is missing.

Page 5, figure 2

-        2B/2D: how was this quantified/what is on the Y-axis: ratio of band intensity?

-        Can you explain why there is a differential effect of Gclc expression to Rot vs. MPP+?

Page 7, figure 4

-        4B/4D/4E: MPTP+AUR is compared to MPTP+DMSO, but how is the difference between MPTP+AUR and CN?

-        Figure description: “Data are presented as means +- SD of three experiments” and “n=10/group”.  Can you clarify this?

Page 8, figure 5

-        In 5D-5G, MPTP+AUR is also compared to CN. Could you add the comparison in 5C as well?

-        5C: unit of the movement distance is missing.

-        5B/5C: what does a decrease in movement mean? Is it

Page 10

-        “In the current study, we tested AUR at a concentration of 1 uM.” Why was the concentration that much lower than in the other study?

-        Cell culture: the SH-SY5Y cells are missing here. In general: the motivation for these two cell lines is missing.

-        Where do the substances (AUR, Rot, MPP+) come from? Prepared by yourself, ordered, received from?

Page 11

-        Please combine the three sections “Animal / MPTP-induced PD mouse model / Auraptene injection”

Page 12

-        Statistical analysis: in general throughout the paper (as commented already in some places) it is not always clear what the mean indicates. Is the mean calculated from (technical) replicates, or (independent) repetitions? Please specificy!

Author Response

6th July, 2019

International Journal of Molecular Sciences Editorial Office

Dear Dr. Owen Chen

We would like to thank the reviewers for the detailed comments on our manuscript (ijms-539563). Please find enclosed our revised manuscript, entitled “Auraptene mitigates Parkinson’s disease-like behavior by enhancing mitochondrial respiration and scavenging reactive oxygen species” which we would like to submit for publication in International Journal of Molecular Sciences.

We are grateful to the reviewers and editors for their suggestions. We believe that the changes made in response to these suggestions have strengthened our manuscript, including revising our findings and their interpretation. We have revised our manuscript to answer the point-by-point response to all comments raised by the reviewers. The changed manuscript is noted by blue colored text.

Sincerely yours,

Please address all correspondence to:

Gi Ryang Kweon, M.D, Ph.D

Professor

Department of Biochemistry,

College of Medicine, Chungnam National University

Munhwa-dong, Jungu, Deajeon 301-747, Republic of Korea

Phone: 82-42-580-8226

Fax: 82-42-580-8121

E-mail: mitochondria@cnu.ac.kr

Jun Young Heo, M.D, Ph.D

Associate Professor

Department of Biochemistry,

College of Medicine, Chungnam National University

Munhwa-dong, Jungu, Deajeon 301-747, Korea

Phone: 82-42-580-8351

Fax: 82-42-580-8121

E-mail: junyoung3@gmail.com

Responses are in bold

Reviewer 2

The article is quite well done, the experiments and outcomes are well described and clearly organized in the paper. However, some minor revisions are needed. After repairing these missing details and inaccuracies, the manuscript can be accepted for publication.

Page 3, figure 1

-        What does CN stands for, the control? Please specify.

We thank to the reviewer’s comment. To clarify the meaning of CN, we added CN means control group in figure legend

If CN is indeed the control: figure 1E/1F are % of control, why is the CN bar in 1F not at 100%?

We appreciated to the reviewer. When we check Fig. 1F, graph showed incorrect control value. We intended to represent AUC of OCR as % of control. Therefore, we set a control at 100% and calculated raw data as % of control.

-        Triplicate samples: does it mean the measurements are performed in triple technical replicates? Or was the experiment repeated three times (independently, with new cells)? Please make clear both how many repetitions and how many replicates were used.

We performed three independent experiments (in vivo and in vitro). Each experiment includes technical replicates to gain reliable data using three or more samples. Then we did statistical analysis of overall data.

-        Figure 1D: the Y-axis labels should probably be 150/100/50 (instead of 150/10/50).

We apologize for scale error of Y-axis in Fig. 1D. We labeled 150/100/50 on the graph. 

-        The neuroprotective effect of AUR seems to be stronger after Rot treatment, compared to after MPP+ treatment (according to OCR results, …). Can you give an explanation why this could be the case?

In response to the reviewer’s comment, we analyzed the increase range of basal OCR. Basal OCR AUC showed 24.1% higher in AUR+Rot treated group than Rot only treated group. And, 31.03% higher in AUR+MPP+ treated group than MPP+ only treated group. From these results, we think the effect of AUR on neuroprotection via modulation of mitochondrial respiration as analyzed by increase range of basal OCR is similar in these neurotoxins.

-        Figure 1F: the AUR-treated groups show a higher bacal OCR than the control group, is it significantly higher and does it have any consequences? Can you explain that?

As the reviewer’s comment, we observed AUR pre-treated and MPP+ treated cells showed higher basal OCR than control. However, there is no significant difference in behavior test such as open-field test and vertical grid test between control and AUR-treated group (Fig. 5). These result suggest that AUR could increase OCR of dopaminergic neuron in the presence of MPP+ and it is consequently sufficient to improve MPTP –induced PD-like behavior to normal level. But, additive beneficial effect on behavior or hypermobility were not found. We added the following setences in second paragraph of discussion section. ‘In the AUR-pretreated and MPP+ -treated group, basal OCR was higher than that of control. However, there is no significant difference in behavioral test such as open-field test and vertical grid test between control and AUR-treated group (Fig. 5). These result suggest that AUR increases OCR of dopaminergic neuron in the presence of MPP+ and it is consequently sufficient to improve MPTP –induced PD-like behavior to normal level. But, additive beneficial effect on behavior or hypermobility were not found. We described this in discussion section paragraph 2.

Page 4

-        NRF2 is mentioned to be involved in regulation of antioxidant-related genes, how? Does it activate/suppress?

In response to the reviewer’s comment, we added reference to result section. NRF2 upregulate antioxidant-related genes [7].

Similarly: “These results indicate that AUR treatment induces NRF2 protein expression in cells.” But what is the consequences of this increased protein expression?

As you see the figure 2 E-F, most of the antioxidant-related genes were upregulated. NRF2 upregulate these genes transcription [7].

-        Here the SH-SY5Y is mentioned for the first time. Why those cells, how do they relate/are they different from the SN4741 cells?

In general: the motivation for the choice of those two cell lines is missing.

In response to the reviewer’s comment, we decided to discard description of experimental result performed with SH-SY5Y. The Supplementary figure 4, the experiment with SH-SY5Y causes some confusion and is beside the point in our manuscript. SH-SY5Y cell line is known to express not only dopaminergic neuron but also cortical neuron and it has a feature of tumor because it is a neuroblastoma cell line. Therefore, we chose SN4741 mouse embryonic dopaminergic cells to validate AUR effect on the neurotoxin- induced PD- like behavior and ROS. Although it shows the increase of NRF2 expression in SH-SY5Y-treated AUR+Rot compared to Rot group, it is better to remove the figure to help to understand our manuscript without confusion.

Page 5, figure 2

-        2B/2D: how was this quantified/what is on the Y-axis: ratio of band intensity?

As a-tubulin is used as an internal control, NRF2 band intensity was normalized to a-tubulin. Because a-tubulin protein level is slightly different in each lane, we measured NRF2/a-tubulin ratio instead of presenting NRF2 intensity only.

-        Can you explain why there is a differential effect of Gclc expression to Rot vs. MPP+?

In response to reviewer’s comments, as we demonstrated in Fig. 1C-F, rotenone showed higher inhibitory effect on mitochondrial respiration of SN4741 cells than MPP+, although both rotenone and MPP+ targets complex I. Therefore, rotenone possibly causes higher ROS level than MPP+. We pretreated AUR for 1 hour before treatment of neurotoxin to induce antioxidant enzyme expression. It is reported that AUR induces hepatic GST activities via the Nrf2/ARE mechanism in mice [8]. However, increased ROS could offset against Gclc induction in rotenone- treated cells. We added this to result section on page 8.  

Page 7, figure 4

-        4B/4D/4E: MPTP+AUR is compared to MPTP+DMSO, but how is the difference between MPTP+AUR and CN?

We thank to the reviewer’s comment. We performed statistical analysis and added significance difference between groups using asterisk. 

-        Figure description: “Data are presented as means +- SD of three experiments” and “n=10/group”.  Can you clarify this?

 In response to the reviewer’s comment, brain slice of mice from SN was obtained through -2.60 mm to -3.90 mm relative to bregma and slices from striatum was obtained through -1.08 mm to -0.84 mm relative to bregma after finishing behavior test. Mice number of each group is 4-5. We picked 10 slices of each area of brain and conducted immunohistochemistry. And we repeated overall experiment in three times independently. 

Page 8, figure 5

-        In 5D-5G, MPTP+AUR is also compared to CN. Could you add the comparison in 5C as well?

In response to reviewer’s comments, we added the comparison of MPTP+AUR and CN in 5C by performing statistical analysis.

-        5C: unit of the movement distance is missing.

In response the comment, we added the unit “cm”.

-        5B/5C: what does a decrease in movement mean? Is it

In response to the reviewer’s comment, as we mentioned in the figure legend, figure 5B means the tracking which shows the movement of mouse in open-field test and the tracking were quantified in figure 5C. You can see the decrease of the MPTP+DMSO-treated mice’s movement compared to CN and MPTP+AUR group.

Page 10

- “In the current study, we tested AUR at a concentration of 1 uM.” Why was the concentration that much lower than in the other study?

In response to the reviewer’s comment, cancer cell metabolism is different from normal cells especially dopaminergic neuronal cell, depending on anaerobic glycolysis to rapidly obtain ATP rather than mitochondrial respiration. We found that unlike 100uM AUR inhibit basal OCR in cancer cell [9], lower concentration of AUR (1uM) significantly increased basal OCR of SN4741 cells without affecting cell viability as confirmed by sulforhodamine B assay. Our strategy is to enhance mitochondrial respiration of dopaminergic neuron in the presence of neurotoxin, because reduction of mitochondrial respiration by genetic mutation and environmental toxin leads to dopaminergic neuronal death in Parkinson’s disease.

-        Cell culture: the SH-SY5Y cells are missing here. In general: the motivation for these two cell lines is missing.

We appreciated the reviewer’s comment. As we mentioned above, we decided to discard description of experimental result performed with SH-SY5Y. SH-SY5Y cell line is known to express not only dopaminergic neuron but also cortical neuron and it has a feature of tumor because it is a neuroblastoma cell line. Therefore, we chose SN4741 mouse embryonic dopaminergic cells to validate AUR effect on the neurotoxin- induced PD- like behavior and ROS.

-        Where do the substances (AUR, Rot, MPP+) come from? Prepared by yourself, ordered, received from?

In response to the reviewer’s comment, we described the information. AUR, rotenone and MPP+ powder was purchased from SIGMA Aldrich.

Page 11

-        Please combine the three sections “Animal / MPTP-induced PD mouse model / Auraptene injection”

According to the reviewer’s suggestion, we combined the methods.

Page 12

-        Statistical analysis: in general throughout the paper (as commented already in some places) it is not always clear what the mean indicates. Is the mean calculated from (technical) replicates, or (independent) repetitions? Please specificy

We performed three independent experiments. Each experiment includes biological replicates using at least three or more samples. Then we proceeded statistical analysis with overall data.

References

1.      Jackson-Lewis, V.; Przedborski, S., Protocol for the MPTP mouse model of Parkinson's disease. Nature protocols 2007, 2, (1), 141-51.

2.      Kim, S. J.; Ryu, M. J.; Han, J.; Jang, Y.; Lee, M. J.; Ju, X.; Ryu, I.; Lee, Y. L.; Oh, E.; Chung, W.; Heo, J. Y.; Kweon, G. R., Non-cell autonomous modulation of tyrosine hydroxylase by HMGB1 released from astrocytes in an acute MPTP-induced Parkinsonian mouse model. Laboratory Investigation 2019.

3.      Dal Ben, M.; Bongiovanni, R.; Tuniz, S.; Fioriti, E.; Tiribelli, C.; Moretti, R.; Gazzin, S., Earliest Mechanisms of Dopaminergic Neurons Sufferance in a Novel Slow Progressing Ex Vivo Model of Parkinson Disease in Rat Organotypic Cultures of Substantia Nigra. International Journal of Molecular Sciences 2019, 20, (9), 2224.

4.      Okuyama, S.; Semba, T.; Toyoda, N.; Epifano, F.; Genovese, S.; Fiorito, S.; Taddeo, V. A.; Sawamoto, A.; Nakajima, M.; Furukawa, Y., Auraptene and Other Prenyloxyphenylpropanoids Suppress Microglial Activation and Dopaminergic Neuronal Cell Death in a Lipopolysaccharide-Induced Model of Parkinson's Disease. Int J Mol Sci 2016, 17, (10).

5.      Dias, V.; Junn, E.; Mouradian, M. M., The role of oxidative stress in Parkinson's disease. J Parkinsons Dis 2013, 3, (4), 461-91.

6.      Meredith, G. E.; Rademacher, D. J., MPTP mouse models of Parkinson's disease: an update. J Parkinsons Dis 2011, 1, (1), 19-33.

7.      Claudia, T.; Christine, C. I. I.; A., T. D., Transcriptional Regulation by Nrf2. Antioxidants & Redox Signaling 2018, 29, (17), 1727-1745.

8.      Prince, M.; Li, Y.; Childers, A.; Itoh, K.; Yamamoto, M.; Kleiner, H. E., Comparison of citrus coumarins on carcinogen-detoxifying enzymes in Nrf2 knockout mice. Toxicol Lett 2009, 185, (3), 180-6.

9.      Jang, Y.; Han, J.; Kim, S. J.; Kim, J.; Lee, M. J.; Jeong, S.; Ryu, M. J.; Seo, K. S.; Choi, S. Y.; Shong, M.; Lim, K.; Heo, J. Y.; Kweon, G. R., Suppression of mitochondrial respiration with auraptene inhibits the progression of renal cell carcinoma: involvement of HIF-1alpha degradation. Oncotarget 2015, 6, (35), 38127-38.

Round 2

Reviewer 1 Report

The revised version of the manuscript of Jang et al. clearly fulfills all the concerns and suggestion that were mentioned in the first version. The Methods and the Results sections are well-described and organized. The final version of the manuscript was improved and clarified.